# Fine-tuning Vision Foundation Models for Multi-Modal Prostate MR Sequence Classification

**Stefan Denner**[1]                                                                 STEFAN.DENNER@DKFZ-HEIDELBERG.DE
**Bálint Kovács**[1]
**David Zimmerer**[1]
**Deepa Krishnaswamy**[2]
**Dimitrios Bounias**[1]
**Raphael Stock**[1]
**Markus Bujotzek**[1]
**Fergus Imrie**[3]
**Andrey Fedorov**[2]
**Klaus H. Maier-Hein**[1]

[1] *Division of Medical Image Computing, German Cancer Research Center, Heidelberg, Germany*

[2] *Brigham and Women's Hospital, Boston, MA, USA*

[3] *Department of Statistics, University of Oxford, Oxford, UK*

**Editors:** Under Review for MIDL 2025

## Abstract

Assigning MRI sequence types is essential yet remains a tedious, manual step in prostate imaging workflows. Current automated approaches relying solely on images or DICOM metadata often struggle with protocol variability and metadata inaccuracies, limiting their generalizability. We propose fine-tuning vision foundation models within different fusion strategies integrating image and metadata. We achieve state-of-the-art F1-score of 1.00 and 0.98 on internal and external test sets, respectively, demonstrating robust generalization.

**Keywords:** MRI, Prostate Cancer, Sequence Classification, Vision Foundation Models

## 1. Introduction

Magnetic resonance imaging (MRI) plays a crucial role in tissue characterization by providing complementary information through multiple sequences with distinct tissue contrasts, making it particularly valuable in the diagnosis of prostate cancer (PCa) and the guidance of subsequent interventions (Turkbey and Choyke, 2018). The PI-RADS guidelines (Weinreb et al., 2016; Turkbey et al., 2019) recommend acquiring prostate MRI in a multiparametric fashion, including T2-weighted (T2w), diffusion-weighted imaging (DWI), and dynamic contrast-enhanced (DCE) sequences. However, machine learning algorithms often require only a subset of the available prostate MRI scans (Bhattacharya et al., 2022), including primary sequences or derived images like the apparent diffusion coefficient (ADC) map. In this context, manual data curation remains a tedious and time-consuming task.

Automatic methods utilizing either DICOM metadata (Gauriau et al., 2020; Cluceru et al., 2023) or imaging data (Kasmanoff et al., 2023; Salome et al., 2023) have been introduced but face limited generalizability due to variability in acquisition protocols and frequent metadata inaccuracies. To enhance robustness, Krishnaswamy et al. (2024) combined

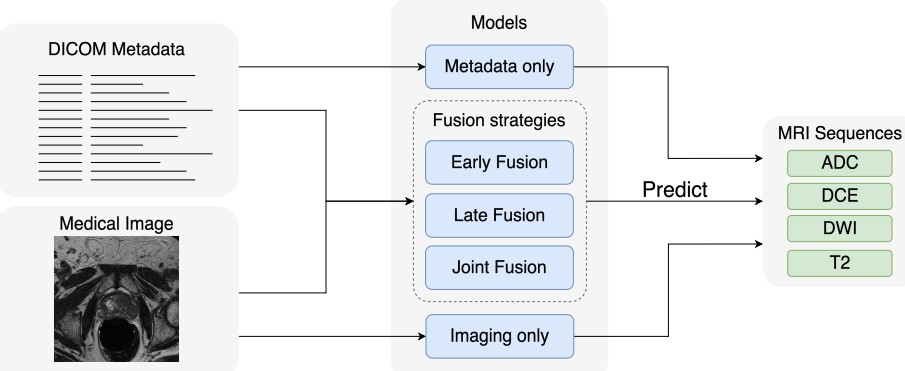

Figure 1:   Overview of the proposed multi-modal MRI sequence classification approach.

image features with DICOM metadata, achieving improved yet still limited generalization due to the necessity of training from scratch on relatively small, annotated prostate MRI datasets.

To overcome the reliance on extensive labeled datasets, we propose fine-tuning pre-trained vision foundation models for multi-modal MRI sequence classification. Specifically, we investigate the impact of different pretraining strategies (supervised, self-supervised, weakly supervised), model sizes, and various fusion strategies integrating image data with metadata. We validate our approach on internal and external datasets, achieving state-of-the-art F1 scores of 1.00 and 0.98, respectively, demonstrating superior generalization capability and robustness.

## 2. Methodology

**Data**   We use the prostate MRI dataset collection introduced by Krishnaswamy et al. (2024), comprising two internal datasets (used for training, validation, and testing) and five external datasets (used exclusively for testing). Following the original preprocessing, we use the extracted center slices from 3D volumes along with the metadata predefined splits.

**Methods**   We evaluate multiple vision foundation models across different fusion strategies. Specifically, we fine-tune three pretrained vision encoders: supervised Vision Transformer (ViT) (Dosovitskiy et al., 2020), self-supervised DINOv2 (Oquab et al., 2023) (both pretrained on natural images), and weakly-supervised BiomedCLIP (Zhang et al., 2023), pretrained on scientific biomedical image-text pairs. Additionally, we assess the impact of varying model sizes.

As baselines, we include image-only and metadata-only approaches. To systematically study multi-modal fusion, we adopt the strategies described in Imrie et al. (2025): (1) Early fusion concatenates extracted image features from the vision encoders with metadata before classification via a multilayer perceptron (MLP); (2) Late fusion ensembles predictions from separate image-only and metadata-only models (using Autoprognosis (Imrie et al., 2025) for metadata); and (3) Joint fusion concatenates image and metadata features and trains the entire model end-to-end. We perform an ablation across all vision encoders and fusion strategies. Full training details are provided in appendix A.

Table 1: F1 scores for our proposed multi-modal MRI sequence classification approach. Avg. best in bold. †= Results taken from (Krishnaswamy et al., 2024)

| | | | | Internal | | | | | External | | |
|---|---|---|---|---|---|---|---|---|---|---|---|
| | | ADC | DCE | DWI | T2w | **Avg.** | ADC | DCE | DWI | T2w | **Avg.** |
| | Images † | 0.99 | 0.99 | 0.99 | 0.99 | 0.99 | 0.99 | 0.89 | 0.59 | 0.93 | 0.85 |
| | Metadata † | 1.00 | 1.00 | 1.00 | 1.00 | 1.00 | 0.91 | 1.00 | 0.60 | 0.98 | 0.87 |
| | Images + metadata † | 0.99 | 0.99 | 1.00 | 1.00 | 1.00 | 0.99 | 0.99 | 0.72 | 0.98 | 0.92 |
| | Autoprognosis | 1.00 | 1.00 | 1.00 | 1.00 | 1.00 | 0.95 | 0.99 | 0.14 | 0.94 | 0.76 |
| **Imaging only** | ViT-B/16 | 1.00 | 1.00 | 1.00 | 1.00 | 1.00 | 0.96 | 0.94 | 0.82 | 0.93 | 0.91 |
| | ViT-L/16 | 1.00 | 1.00 | 1.00 | 1.00 | 1.00 | 0.98 | 0.92 | 0.88 | 0.87 | 0.91 |
| | DINOv2 ViT-S/14 | 1.00 | 1.00 | 1.00 | 1.00 | 1.00 | 0.93 | 0.94 | 0.88 | 0.93 | 0.92 |
| | DINOv2 ViT-B/14 | 1.00 | 1.00 | 1.00 | 1.00 | 1.00 | 0.98 | 0.91 | 0.92 | 0.85 | 0.92 |
| | DINOv2 ViT-L/14 | 1.00 | 1.00 | 1.00 | 1.00 | 1.00 | 0.94 | 0.92 | 0.90 | 0.90 | 0.91 |
| | BiomedCLIP ViT-B/16 | 1.00 | 1.00 | 1.00 | 1.00 | 1.00 | 0.98 | 0.96 | 0.93 | 0.92 | 0.95 |
| **Early Fusion** | ViT-B/16 | 1.00 | 1.00 | 1.00 | 1.00 | 1.00 | 0.96 | 0.96 | 0.82 | 0.97 | 0.93 |
| | ViT-L/16 | 1.00 | 1.00 | 1.00 | 1.00 | 1.00 | 0.81 | 0.97 | 0.89 | 0.67 | 0.84 |
| | DINOv2 ViT-S/14 | 0.96 | 1.00 | 1.00 | 0.97 | 0.98 | 0.87 | 0.99 | 0.87 | 0.88 | 0.90 |
| | DINOv2 ViT-B/14 | 1.00 | 1.00 | 1.00 | 1.00 | 1.00 | 0.98 | 1.00 | 0.90 | 0.99 | 0.97 |
| | DINOv2 ViT-L/14 | 1.00 | 1.00 | 0.98 | 0.96 | 0.98 | 0.68 | 1.00 | 0.84 | 0.35 | 0.72 |
| | BiomedCLIP ViT-B/16 | 1.00 | 1.00 | 1.00 | 1.00 | 1.00 | 0.98 | 1.00 | 0.88 | 0.98 | 0.96 |
| **Late fusion** | ViT-B/16 | 1.00 | 1.00 | 1.00 | 1.00 | 1.00 | 0.98 | 0.99 | 0.89 | 0.99 | 0.96 |
| | ViT-L/16 | 1.00 | 1.00 | 1.00 | 1.00 | 1.00 | 0.98 | 0.98 | 0.92 | 0.97 | 0.96 |
| | DINOv2 ViT-S/14 | 1.00 | 1.00 | 1.00 | 1.00 | 1.00 | 0.95 | 0.98 | 0.81 | 0.98 | 0.93 |
| | DINOv2 ViT-B/14 | 1.00 | 1.00 | 1.00 | 1.00 | 1.00 | 0.97 | 0.94 | 0.88 | 0.89 | 0.92 |
| | DINOv2 ViT-L/14 | 1.00 | 1.00 | 1.00 | 1.00 | 1.00 | 0.95 | 0.98 | 0.85 | 0.97 | 0.94 |
| | BiomedCLIP ViT-B/16 | 1.00 | 1.00 | 1.00 | 1.00 | 1.00 | 0.97 | 0.98 | 0.91 | 0.97 | 0.96 |
| **Joint fusion** | ViT-B/16 | 1.00 | 1.00 | 1.00 | 1.00 | 1.00 | 0.97 | 0.88 | 0.91 | 0.57 | 0.83 |
| | ViT-L/16 | 1.00 | 1.00 | 1.00 | 1.00 | 1.00 | 0.98 | 0.95 | 0.94 | 0.86 | 0.93 |
| | DINOv2 ViT-S/14 | 1.00 | 1.00 | 1.00 | 1.00 | 1.00 | 0.96 | 1.00 | 0.86 | 1.00 | 0.95 |
| | DINOv2 ViT-B/14 | 1.00 | 1.00 | 1.00 | 1.00 | 1.00 | 0.98 | 1.00 | 0.93 | 0.99 | **0.98** |
| | DINOv2 ViT-L/14 | 1.00 | 1.00 | 1.00 | 1.00 | 1.00 | 0.98 | 1.00 | 0.93 | 0.99 | 0.97 |
| | BiomedCLIP ViT-B/16 | 1.00 | 1.00 | 1.00 | 1.00 | 1.00 | 0.98 | 1.00 | 0.93 | 1.00 | **0.98** |

## 3. Results and Discussion

Our results (table 1) show perfect performance (F1 = 1.00) on the internal test dataset for image-only, late fusion, and joint fusion strategies across all evaluated sequence types. On the external dataset, joint fusion using either BiomedCLIP ViT-B/16 or DINOv2 ViT-B/14 achieves the best overall performance (F1 = 0.98). Specifically, T2w and DCE sequences consistently reach perfect scores (F1 = 1.00), while performance slightly drops for ADC (F1 = 0.98) and notably for DWI (F1 = 0.93). Notably, DWI classification substantially benefits from image information compared to metadata alone (F1 = 0.93 vs. 0.60), underscoring the importance of visual data for this sequence. Across all strategies, BiomedCLIP ViT-B/16 consistently yields the highest overall performance, likely due to its extensive weakly-supervised pretraining on biomedical image-text pairs. Additionally, self-supervised DINOv2 models marginally outperform supervised ViT models on average. Overall, our approach demonstrates that fine-tuned vision foundation models with joint image-metadata fusion effectively generalize across datasets, significantly reducing dependence on large-scale annotated datasets.

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

Table 2: Dataset collection. Number of MR series (patients in parentheses) included for the analysis. ERC = endorectal coil used, †=multiple manufacturers, ‡=multiple magnetic field strengths. Table adapted with minor modifications from (Krishnaswamy et al., 2024)

|  | Dataset | ERC | T2W | DWI | ADC | DCE | Train | Val | Test |
|---|---|---|---|---|---|---|---|---|---|
| Internal | QIN-Prostate-Repeatability (Fedorov et al., 2018) | ✓ | 30 (15) | 30 (15) | 30 (15) | 30 (15) | ✓ | ✓ | ✓ |
| | ProstateX† (Litjens et al., 2014, 2017) | – | 431 (346) | 357 (346) | 356 (346) | 15456 (346) | ✓ | ✓ | ✓ |
| External | Prostate-MRI (Choyke et al., 2016) | ✓ | 26 (26) | 52 (26) | – | 51 (26) | – | – | ✓ |
| | Prostate-3T† (Litjens et al., 2016) | – | 64 (64) | – | – | – | – | – | ✓ |
| | Prostate-Diagnosis (Bloch et al., 2015) | ✓ | 93 (91) | – | – | – | – | – | ✓ |
| | Prostate-MRI-US-Biopsy†‡ (Natarajan et al., 2013) (Sonn et al., 2013) | ✓ | 958 (792) | 110 (108) | 1019 (836) | – | – | – | ✓ |
| | Prostate-Fused-MRI-Pathology (Singanamalli et al., 2016) (Madabhushi and Feldman, 2016) | ✓ | 46 (27) | 13 (12) | 12 (12) | 102 (28) | – | – | ✓ |

## Appendix A. Trainings details

We fine-tune the vision foundation models using a two step approach. We first freeze the backbone and only fine-tune the last layer with a learning rate of 0.0001 for 5 epochs. Then, we unfreeze the whole model and continue fine-tuning with a learning rate of 1e-6 until convergence of the weighted cross entropy validation loss. We ensemble the four cross-fold validation trained models to generate the final prediction. For autoprognosis (Imrie et al., 2023), used in late fusion, we utilize the default configurations with the number of folds set to four.

