# OpenReview forum: "Fine-tuning Vision Foundation Models for Multi-Modal Prostate MR Sequence Classification"
_MIDL.io/2025/Short_Papers — MIDL 2025 - Short Papers_

### Official Review · Reviewer_AvaV · 2025-04-23

**Rating:** 4
**Confidence:** 5

**Summary:**

The authors compare multi-modality learning for classifying MRI sequence types. They compare different fusion strategies, model architectures and evaluate on an external dataset.

**Strengths:**

The methodology and results are impressive, and a robust classification would be very useful in clinical practice.

**Weaknesses:**

I'm wondering if the traditional classification categories really make sense in the scope of sequence classification. Based on the vendor or just the sequence settings, T2w images can look quite different: is it really reasonable to classify all of them as T2w? Perhaps a more in-depth analysis be more useful. I have the same concern when it comes to ADC classification without regarding the b-values used. This essentially aggregates dozens of DICOM tags into four categories which might mask some very important differences. If this method is used to classify images as ADC and then these images are used directly by some deep learning algorithm it might lead to sub-optimal results. This is just some food for thought for the authors.

---

### Decision · Program_Chairs · 2025-05-01

Accept